# Dict-TTS: Learning to Pronounce with Prior Dictionary Knowledge for Text-to-Speech

**Ziyue Jiang**[*]
Zhejiang University
ziyuejiang@zju.edu.cn

**Zhe Su**[*]
Zhejiang University
suzhesz00@gmail.com

**Zhou Zhao**[†]
Zhejiang University
zhaozhou@zju.edu.cn

**Qian Yang**
Zhejiang University
qyang1021@foxmail.com

**Yi Ren**
Bytedance AI Lab
ren.yi@bytedance.com

**Jinglin Liu**
Zhejiang University
jinglinliu@zju.edu.cn

**Zhenhui Ye**
Zhejiang University
zhenhuiye@zju.edu.cn

## Abstract

Polyphone disambiguation aims to capture accurate pronunciation knowledge from natural text sequences for reliable Text-to-speech (TTS) systems. However, previous approaches require substantial annotated training data and additional efforts from language experts, making it difficult to extend high-quality neural TTS systems to out-of-domain daily conversations and countless languages worldwide. This paper tackles the polyphone disambiguation problem from a concise and novel perspective: we propose Dict-TTS, a semantic-aware generative text-to-speech model with an online website dictionary (the existing prior information in the natural language). Specifically, we design a semantics-to-pronunciation attention (S2PA) module to match the semantic patterns between the input text sequence and the prior semantics in the dictionary and obtain the corresponding pronunciations; The S2PA module can be easily trained with the end-to-end TTS model without any annotated phoneme labels. Experimental results in three languages show that our model outperforms several strong baseline models in terms of pronunciation accuracy and improves the prosody modeling of TTS systems. Further extensive analyses demonstrate that each design in Dict-TTS is effective. The code is available at https://github.com/Zain-Jiang/Dict-TTS.

## 1 Introduction

Capturing the pronunciations from raw texts is challenging for end-to-end text-to-speech (TTS) systems [2, 31, 42, 45, 55, 27, 14, 41, 22, 21], since there are full of words that are not covered by general pronunciation rules [4, 24, 50]. Therefore, polyphone[3] disambiguation (one of the biggest challenges in converting texts into phonemes [37, 60, 46]) plays an important role in the construction of high-quality neural TTS systems [18, 38]. However, since the exact pronunciation of a polyphone must be inferred based on its semantic contexts, current solutions still face several challenges: 1) the rule-based approaches [64, 19] with limited linguistic knowledge or the neural models [60, 38, 46]

---

[*]Equal contribution.

[†]Corresponding author

[3]Polyphones are characters having more than one phonetic value. See Appendix D for further details.

36th Conference on Neural Information Processing Systems (NeurIPS 2022).

| Character | Pronunciation | Definitions | | Usages |
|---|---|---|---|---|

乐:
<lè> 欢喜，快活；使人快乐的事情... | 快乐。乐融融。其乐无穷。乐观。乐天。取乐。逗乐。快乐...
<yuè> 声音，成调的声音。或姓氏... | 音乐。声乐。乐池。乐音。乐曲（①音乐与歌曲；②伴奏...
<yào> 喜好、欣赏。用于文言文... | 知者乐水，仁者乐山。
<lào> 地名用字。 | 河北省乐亭、山东省乐陵。

Figure 1: The illustration of the dictionary entry that contains information on character's or word's definitions, usages, and pronunciations. For example, in the Chinese sentence "快乐的人们", the pronunciation of the polyphone "乐" should be inferred based on its semantic contexts.

trained on limited data suffer from significant performance degradation on out-of-domain text datasets; 2) the neural network-based models [6, 5, 38] learn the grapheme-to-phoneme (G2P) mapping in an end-to-end manner without explicit semantics modeling, which hinders their pronunciation accuracy in real-life applications. 3) based on the above two points, a reliable polyphone disambiguation module is usually based on a combination of hand-crafted rules, structured G2P-oriented lexicons, and neural models [16], which requires substantial phonemes labels and external knowledge from language experts.

Unlike the previous rule-based or neural network-based approaches, we address the above challenges with the existing prior information worldwide. As shown in Figure 1, an arbitrary dictionary used in daily life can be viewed as a prior knowledge database. Intuitively, it contains valuable prior knowledge for pronunciations in conversations. When one is confused about the acoustic pronunciation of a specific polyphone, he or she will resort to the dictionary website to infer its exact reading based on the semantic context. We imitate this scenario in our architecture design and propose Dict-TTS, an unsupervised polyphone disambiguation framework, which explicitly consults the online dictionary to identify the correct semantic meanings and acoustic pronunciations of polyphones. Specifically,

- To explicitly learn the semantics-to-pronunciation mapping, we adopt a semantic encoder to obtain the semantic contexts of the input text and utilize a semantics-to-pronunciation attention (S2PA) module to search the matched semantic patterns in the dictionary so as to find the correct pronunciations. We also use the retrieved semantic information as auxiliary information for prosody modeling of the TTS model.

- To perform polyphone disambiguation without phoneme labels, we combine our S2PA module into end-to-end TTS systems' training and inference processes. Different from current neural polyphone disambiguation models, our module can be trained with the guidance of mel-spectrogram reconstruction loss in a fully end-to-end manner, which significantly reduces the cost of building such a system.

To demonstrate the generalization ability of our Dict-TTS, we perform experiments on three datasets, including a standard Mandarin dataset [3], a Japanese corpus [47], and a Cantonese dataset [1]. Experiments on these datasets show that Dict-TTS outperforms other state-of-the-art polyphone disambiguation models in pronunciation accuracy and improves the prosody modeling of TTS systems in terms of both subjective and objective evaluation metrics. The pronunciation accuracy of Dict-TTS is further improved by being pre-trained on a large-scaled automatic speech recognition (ASR) dataset. The main contributions of this work are summarized as follows:

- We incorporate the online dictionary into TTS systems and propose a semantic-aware method for polyphone disambiguation, which improves the pronunciation accuracy and robustness of end-to-end TTS systems. Moreover, the idea of introducing the prior knowledge worldwide can also inspire other tasks like neural language modeling [26] and sequence labeling [34].

- We propose a novel and general framework for unsupervised polyphone disambiguation in TTS systems, which further enables the efficient pre-training on large-scaled ASR datasets and improves the generalization capacity significantly.

- We also find that the retrieved semantics in the dictionary knowledge can be used as auxiliary information to improve prosody modeling and help the TTS system to generate more expressive speech.

- We further analyze the characteristics of the linguistic encoder based on phoneme and character and provide valuable interpretations about our semantics-to-pronunciation attention module.

## 2 Background

This section describes the background of TTS, grapheme-to-phoneme (G2P) pipeline, and their relations with polyphone disambiguation. We also review the existing works that aim at semantic-aware polyphone disambiguation and analyze their advantages and disadvantages.

**Text-to-Speech**  Text-to-speech (TTS) models [55, 2, 30, 31, 42, 27, 41, 32] first generate mel-spectrogram from text and then synthesize speech waveform from the generated mel-spectrogram using a separately pre-trained vocoder [52, 29, 20], or directly generate waveform from text in an end-to-end manner [40, 13, 28]. The frontend model of end-to-end TTS system should tackle one important task, i.e., polyphone disambiguation [65]. Properly mapping the grapheme sequence into phoneme sequence requires the linguistic encoder to capture the empirical pronunciation rules in daily conversations. However, it is extremely difficult for the linguistic encoder to learn all the pronunciation rules necessary to produce speech in an end-to-end manner, which results in inevitable mispronunciations in the generated speech. To alleviate this problem, robust TTS models usually convert the text sequence into the phoneme sequence with open-source grapheme-to-phoneme pipelines and predict the mel-spectrogram from the phoneme sequence. However, the rule-based or neural network-based grapheme-to-phoneme pipelines suffer from significant performance degradation on out-of-domain datasets since it is extremely difficult and costly for them to cover all linguistic knowledge.

**Grapheme-to-Phoneme**  The grapheme-to-phoneme models map grapheme sequence to phoneme sequence to reduce pronunciation errors in modern TTS systems. For logographic languages like Chinese, Japanese and Korean, although the lexicon can cover nearly all the characters, there are full of polyphones that can only be decided according to the semantic context of a character [50]. Thus, polyphone disambiguation is the most important challenge in grapheme-to-phoneme conversions for this kind of languages. Moreover, many alphabetic languages including English and French also have polyphones in daily conversations. Current polyphone disambiguation approaches can be categorized into the rule-based approach [64, 19] and the data-driven approach [43, 6, 5, 38]. The rule-based G2P method is based on a combination of hand-crafted rules and structured G2P-oriented lexicons, which requires a substantial amount of linguistic knowledge. The data-driven G2P algorithm adopts statistical methods [33] or neural encoder-decoder architecture [58, 5, 48, 60, 38, 46, 54]. However, building a data-driven G2P model requires a large amount of carefully labeled data and substantial linguistic knowledge from language experts, which is extremely costly and laborious.

**Semantic-Aware Polyphone Disambiguation**  Polyphone disambiguation is the core issue for G2P conversion in various languages. The pronunciation of a polyphone is defined by the semantic context information of neighbouring characters [50]. In order to comprehend the semantic meaning in the given sentence for polyphone disambiguation, previous methods [11, 57, 49, 18, 8] have adopted the pre-trained language model [12] to extract semantic features from raw character sequences and predict the pronunciation of polyphones with neural classifiers according to the semantic features. Among them, PnG BERT and Mixed-Phoneme BERT [25, 62] take both phoneme and grapheme as input to train an augmented BERT and use the pre-trained augmented BERT as the TTS encoder. However, these methods still require annotated data to train and can not be incorporated into the TTS training in an end-to-end manner. Although NLR [16] directly injects BERT-derived knowledge into the TTS systems without phoneme labels and successfully reduces pronunciation errors, their method confounds the acoustic and semantic space, which significantly affects the pronunciation accuracy.

## 3 Method

To exploit the prior linguistic knowledge in the online dictionary for TTS systems, we propose Dict-TTS, which explicitly captures the semantic relevance between the input sentences and the dictionary entries for polyphone disambiguation. In this section, we firstly introduce the overall architecture design of Dict-TTS based on PortaSpeech [41]. Then after the comparison between the

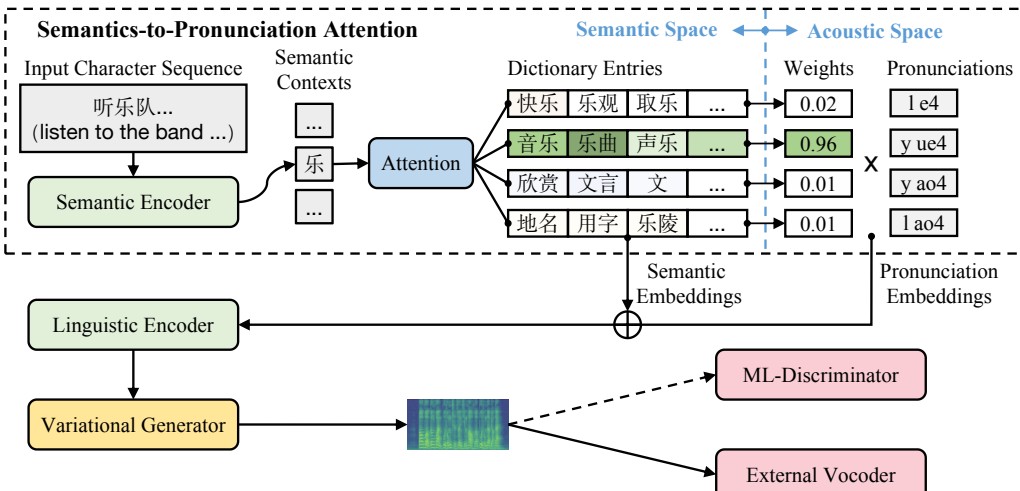

Figure 2: The overall architecture for Dict-TTS. The character "乐" has 4 possible pronunciations coupled with 4 different meanings. The module inside the dashed box is our semantics-to-pronunciation attention (S2PA). The S2PA module measures the semantic similarity between the input character and the corresponding dictionary entries and aggregates the attention weights into pronunciation weights. Then the weighted semantic embeddings and pronunciation embeddings are fed into the linguistic encoder for feature fusion. The semantic and acoustic spaces described in Subsection 3.2 are separated by the blue dashed line. "ML-Discriminator" denotes Multi-Length Discriminator in HiFiSinger [7]. The dashed black line denotes that the operation is only executed in the training phase.

phoneme-based and character-based TTS systems in the acoustic and semantic space, we design a novel semantics-to-pronunciation attention (S2PA) module to learn grapheme-to-phoneme mappings based on the semantic context; In general, Dict-TTS exploits the prior pronunciation knowledge in the online dictionary with the following steps: Firstly, the character sequence is fed into the self-attention based semantic encoder to obtain the semantic representations of the input character sequence, and we utilize a pre-trained cross-lingual language model [10] to extract the semantic context information in the dictionary entries; Secondly, we calculate the most relevant dictionary entries of the input graphemes and obtain the corresponding pronunciation sequence contained in the dictionary entries; Finally, the extracted semantic context information and the pronunciations are fed into the linguistic encoder for feature fusion. We describe these designs in detail in the following subsections.

### 3.1 Model Overview

The overall model architecture of Dict-TTS is shown in Figure 2. Dict-TTS keeps the main structures of PortaSpeech: a Transformer-based linguistic encoder; a VAE-based variational generator with flow-based prior to generate diverse mel-spectrogram. The flow-based post-net in PortaSpeech is replaced with a multi-length discriminator [7] based on random windows of different lengths, which has been proved to improve the naturalness of word pronunciations [59]. However, since there are no phoneme inputs in our scenarios, we replace the linguistic encoder that combines hard word-level alignment and soft phoneme-level alignment with: 1) a semantic encoder that extracts the semantic representations in the grapheme sequence; 2) a semantics-to-pronunciation attention module that matches the semantic patterns between the dictionary entries and the grapheme representations and obtains the corresponding semantic embedding and pronunciation embedding; 3) a linguistic encoder that fuses the semantic embedding and pronunciation embedding.

### 3.2 Comparison between the phoneme-based and character-based TTS systems

In this subsection, we make preliminary analyses about the linguistic encoder of phoneme-based and character-based TTS systems. For simplicity, we describe the following concepts according to the logographic writing system, where a single written character represents a complete grammatical word

or morpheme. These concepts can be extended to alphabetic languages like English by replacing "character" with "word".

**Phoneme-based TTS systems** As is shown in Figure 3, the linguistic encoder of the phoneme-based TTS system takes phoneme sequence $p$ generated by the G2P module as inputs. The main challenge of the phoneme-based linguistic encoder is to comprehend the semantic and syntactic representation $s$ from $p$ and deduce the pitch trajectory, speaking duration, and other acoustic features from $s$ and $p$ to generate the expressive and natural pronunciation hidden state $g$. Since the phoneme sequence $p$ is a combination of the smallest units of sound in speech, it may be ambiguous in terms of semantic meaning, which brings difficulties for the deduction of the representation $s$. For example, "AE1, T, F, ER1, S, T" can be easily classified as "At first", but "W, EH1, DH, ER0" can be classified as "Whether" or "Weather". Homophones like "to", "too", and

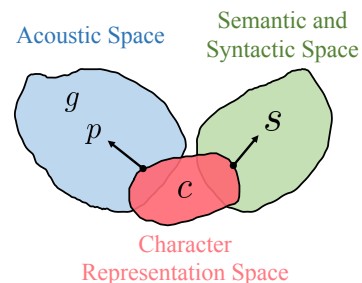

Figure 3: The illustration of representations in the linguistic encoder.

"two" can be converted to the same phoneme sequence "T, UW1", but their local speaking duration and pitch are different. Moreover, the tree-structured syntactic information contained in the word-based input sequence is missing. Since semantic information and syntactic information possess rich intonational features such as pitch accent and phrasing of the input text [59], the ambiguity of $s$ hurts the prosody modeling in phoneme-based TTS systems.

**Character-based TTS systems** The first challenge for a character-based TTS system is to predict the correct phoneme sequence $p$. Unlike the phoneme-based TTS system, the character-based TTS system does not know the phoneme sequence $p$ when characters arrive. Thus, it does not know how to pronounce the words accurately at first. Then the mel-spectrogram reconstruction loss in TTS training would drag the character representation $c$ to the acoustic space. For example, Chinese characters "火" ("fire" in English) and "伙" ("partner" in English) share the same pronunciation ("H UO3") and have different semantic meanings. However, with the guidance of the mel-spectrogram reconstruction loss, their representations distribute according to the acoustic pronunciation, which hinders the semantic comprehension for polyphone disambiguation and prosody modeling.

From the above analyses, it can be seen that the character representations $c$ should locate in the semantic space so that we can easily capture $s$ based on the context, deduce $p$ based on the dictionary and $s$, and finally obtain the natural pronunciation hidden state $g$. The following subsection mainly describes how we achieve the above goals with our semantics-to-pronunciation attention module.

### 3.3 Semantics-to-Pronunciation Attention

As shown in Figure 2, the semantics-to-pronunciation attention (S2PA) module is designed for explicit semantics comprehension and polyphone disambiguation.

**Dictionary Definition** Assuming that we have an dictionary $D$ which contains a sequence of characters $C = [c_1, c_2, ..., c_n]$, where $n$ is the size of characters set in a language[4]. In the dictionary, each character $c_i$ has a sequence of possible pronunciations $p_i = [p_{i,1}, p_{i,2}, ..., p_{i,m}]$ and each pronunciation $p_{i,j}$ has its corresponding dictionary entry $e_{i,j} = [e_{i,j,1}, e_{i,j,2}, ..., e_{i,j,u}] \in E$ (definitions, usages, and translations are merged together as a single characters sequence), where $m$ is the number of the possible pronunciation of $c_i$ and $u$ is the number of characters in the corresponding entry. Note that for polyphones $m > 1$ and for characters that have only one pronunciation $m = 1$.

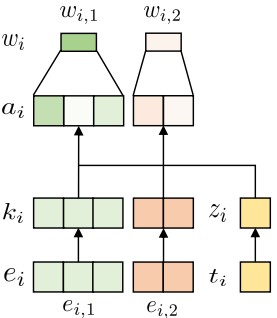

Figure 4: The illustration of semantic pattern matching in S2PA module.

---

[4]The dictionary can be easily downloaded from online websites. See Appendix A.1 for further details

**Semantics Matching**   The goal of our S2PA is to obtain the pronunciation sequence $p$ by measuring the semantic similarity between the input character sequence $t = [t_1, ..., t_l]$ and the corresponding gloss items in the dictionary, where $l$ is the sequence length. As shown in Figure 4, we firstly extract the semantic context information $\mathbf{k}$ of every entry $e$ with a pretrained cross-lingual language model [10] and store $\mathbf{k}$ as the prior dictionary knowledge. Then we use a semantic encoder to obtain the semantic contexts $\mathbf{z}$ from the input character sequence $t$. For each character token $t_i$, its semantic feature vector $\mathbf{z}_i$ is used as the query vector to a semantics-based attention module. Here, attention is used to learn a similarity measure between the semantic feature vector $\mathbf{z}_i$ and $\mathbf{k}_{i,j,k}$:

$$[\mathbf{a}_{i,1,1}, ..., \mathbf{a}_{i,m,u}] = \frac{[\mathbf{k}_{i,1,1}, ..., \mathbf{k}_{i,m,u}] \cdot \mathbf{z}_i^\top}{d} ,\tag{1}$$

where $d$ is the scaling factor and $[\mathbf{a}_{i,1,1}, ..., \mathbf{a}_{i,m,u}]$ denotes the semantic similarity between $t_i$ and each item in $[e_{i,1,1}, e_{i,m,u}]$. The retrieved semantic embeddings $\mathbf{s}'_i$ can be extracted by $\mathbf{s}'_i = softmax([\mathbf{a}_{i,1,1}, ..., \mathbf{a}_{i,m,u}]) \cdot [\mathbf{k}_{i,1,1}, ..., \mathbf{k}_{i,m,u}]$. The rich linguistic information in $\mathbf{s}'_i$ can be used as auxiliary information to improve the naturalness and expressiveness of the generated speeches.

**Polyphone Disambiguation**   The aggregated attention weight $\mathbf{w}_{i,j} = \sum_{k=1}^{u} \mathbf{a}_{i,j,k}$ can be seen as the probability of the pronunciation $p_{i,j}$. However, since a polyphone in a specific sentence has only one correct pronunciation, here we use the Gumbel-Softmax function [23] to sample a differentiable approximation of the most possible pronunciation $p'_i$:

$$w_{i,j} = \frac{\exp\left(\left(\log\left(\mathbf{w}_{i,j}\right) + g_{i,j}\right)/\tau\right)}{\sum_{l=1}^{m} \exp\left(\left(\log\left(\mathbf{w}_{i,l}\right) + g_{i,l}\right)/\tau\right)} ,\tag{2}$$

$$p'_i = \sum_{j=1}^{m} w_{i,j} \cdot p_{i,j} ,\tag{3}$$

where $g_{i,1}, ..., _{i,m}$ are i.i.d samples drawn from Gumbel(0,1) distribution and $\tau$ is the softmax temperature. Then the retrieved pronunciation embeddings $p'_i$ and semantic embeddings $\mathbf{s}'_i$ are then fed into the rest of the linguistic encoder for feature fusion and syntax prediction. Intuitively, our S2PA module can be thought of as an end-to-end method for decomposing the character representation, pronunciation, and semantics with dictionary following the preliminary analyses in Subsection 3.2. Through our S2PA module, the character representation is successfully distributed in the semantic space so that the model can easily deduce the correct pronunciation and semantics based on the lexicon knowledge like human brain.

### 3.4   Training and Pre-training

In training, the S2PA module weights (including character token embeddings and the pronunciation token embeddings) are jointly trained by the reconstruction loss from the TTS decoder. Thus, our Dict-TTS does not require any explicit phoneme labels. In inference, the pronunciations can be specified by feeding the text sequence into the S2PA module to get the predicted pronunciations. Besides, our method is compatible with the predefined rules from language experts by directly adding specific rules to pronunciation weight $w_{i,j}$.

Although the semantics-to-pronunciation mappings can be explicitly learned by the S2PA module, it could be not accurate enough, due to the following reason: the text training data is not large enough (about 10k sentences) in the TTS dataset, leading to relatively inaccurate context comprehension. To improve the semantic comprehension and the generalization capacity for our S2PA module, we propose a pre-training method using low-quality text-speech pairs from large-scaled automatic speech recognition (ASR) datasets. Since our S2PA module can be trained without hand-crafted phoneme labels, it can be easily pre-trained and effectively finetuned to various domains to improve the pronunciation accuracy of the TTS systems.

## 4   Experiments

### 4.1   Experimental Setup

**Datasets**   We evaluate Dict-TTS on three datasets of different sizes, including: 1) Biaobei [3], a Chinese speech corpus consisting of 10,000 sentences (about 12 hours) from a Chinese speaker;

Table 1: The objective and subjective pronunciation accuracy comparisons. PER-O denotes phoneme error rate in the objective evaluation, PER-S denotes phoneme error rate in the subjective evaluation and SER-S denotes sentence error rate in the subjective evaluation.

| Method | Biaobei | | | JSUT | | | Common Voice (HK) | | |
|---|---|---|---|---|---|---|---|---|---|
| | PER-O | PER-S | SER-S | PER-O | PER-S | SER-S | PER-O | PER-S | SER-S |
| Character | - | 3.73% | 30.50% | - | 13.78% | 65.50% | - | 1.89% | 15.50% |
| Phoneme | 2.78% | 1.14% | 7.00% | **1.55%** | **0.92%** | **4.25%** | - | 1.45% | 10.25% |
| Dict-TTS | **2.12%** | **1.08%** | **6.50%** | 3.73% | 2.57% | 22.75% | - | **1.23%** | **9.75%** |

2) JSUT [47], a Japanese speech corpus containing reading-style speeches from a Japanese female speaker (we use the basic5000 subset that contains 5,000 daily-use sentences). 3) Common Voice (HK) [1], a Cantonese speech corpus that contains 125 hours of speeches from 2,869 speakers (We use the 102 hours of the validated speeches). For each of the three datasets, we randomly sample 400 samples for validation and 400 samples for testing. We randomly choose 50 samples in the test set for subjective audio quality and prosody evaluation and use all testing samples for other evaluations. The ground truth mel-spectrograms are generated from the raw waveform with the frame size 1024 and the hop size 256. For computational efficiency, we firstly use the pre-trained XLM-R [10] model to extract the semantic representations from the raw text of the whole dictionary and record them in the disk. We then load the mini-batch along with the pre-constructed dictionary representation during training and testing.

**Implementation Details** Our Dict-TTS consists of a S2PA module, an encoder, a variational generator and a post-net. The encoder consists of multiple feed-forward Transformer blocks [42] with relative position encoding [44] following Glow-TTS [27]. The encoder and decoder in variational generator are 2D-convolution networks following PortaSpeech [41]. We replace the post-net in PortaSpeech with a multi-length discriminator [7], which has been proved to improve the naturalness of word pronunciations [59]. We add more detailed model configurations in Appendix B.1, B.2. We train Dict-TTS on 1 NVIDIA 3080Ti GPU, with batch size of 40 sentences on each GPU. We use the Adam optimizer with $\beta_1 = 0.9$, $\beta_2 = 0.98$, $\epsilon = 10^{-9}$ and follow the same learning rate schedule in [53]. The softmax temperature $\tau$ is initialized and annealed using the schedule in [23]. It takes 320k steps for training until convergence. The predicted mel-spectrograms are transformed into audio samples using pre-trained HiFi-GAN [29][5].

## 4.2 Results of Pronunciation Accuracy

We compare the pronunciation accuracy of our Dict-TTS with other systems, including 1) character-based systems, where we directly feed character into the linguistic encoder; 2) Phoneme-based systems, where we convert the text sequence to the phoneme sequence [55, 45, 27] with most popular open-source grapheme-to-phoneme tools[6]. We measure objective phoneme error rate (PER-O), subjective phoneme error rate (PER-S), and subjective sentence error rate (SER-S) in the evaluations. The phoneme labels in the objective PER evaluation are from the corresponding dataset (since the Common

Table 2: The objective and subjective pronunciation accuracy comparisons in the Biaobei dataset.

| Method | PER-O | PER-S | SER-S |
|---|---|---|---|
| Character | - | 3.73% | 30.50% |
| BERT Embedding [15] | - | 4.03% | 38.75% |
| NLR [16] | - | 2.98% | 26.50% |
| Phoneme (G2PM) | 3.95% | 1.39% | 10.50% |
| Phoneme (pypinyin) | 2.78% | 1.14% | 7.00% |
| Dict-TTS | 2.12% | 1.08% | 6.50% |
| Dict-TTS (pre-trained) | **1.54%** | **0.79%** | **4.25%** |

Voice (HK) dataset does not have phoneme labels, we only evaluate the subjective metrics). In the subjective evaluations, each audio in the test set is listened by at least 4 language experts. We ask them to write down the mispronounced phonemes and discuss them with each other until a

---

[5]`https://github.com/jik876/hifi-gan`

[6]We use *pypinyin* in the Biaobei dataset, *pyopenjtalk* in the JSUT dataset and *pycantonese* in the Common Voice (HK) dataset. More detailed information can be found in Appendix B.4

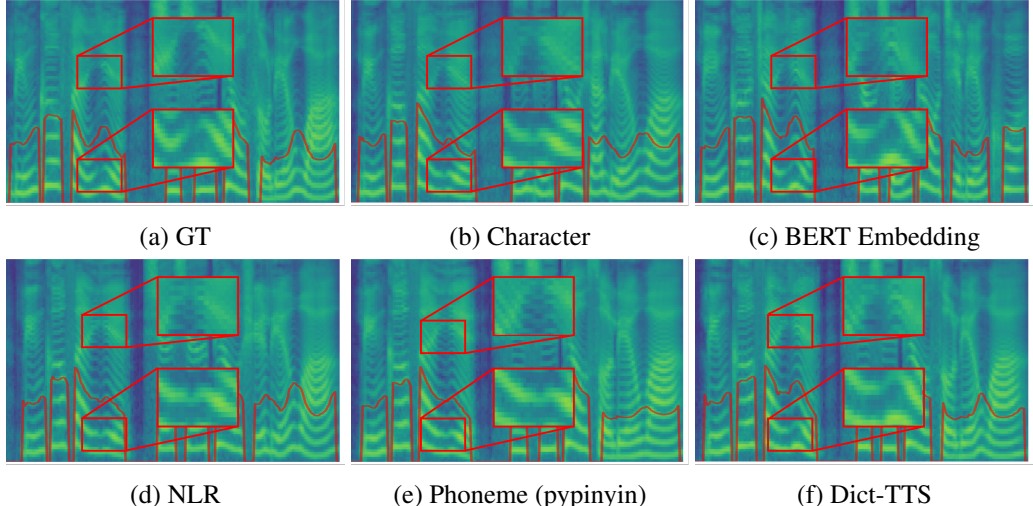

|     |     |     |
| :-: | :-: | :-: |
| (a) GT | (b) Character | (c) BERT Embedding |
| (d) NLR | (e) Phoneme (pypinyin) | (f) Dict-TTS |

Figure 5: Visualizations of the ground-truth and generated mel-spectrograms by different types of linguistic encoders. The corresponding text is "菱歌泛夜，嬉嬉钓叟莲娃", which means "The man picking water chestnut sings at night. The old man fishing and the girl picking lotus are laughing".

conclusion is reached. Note that for SER-S, the error rate is calculated in sentence level (e.g., a sentence with multiple errors will be counted only once). More details about these evaluations can be found in Appendix B.3. The results are shown in Table 1. It can be seen that Dict-TTS greatly surpasses the character-based baseline in three languages. Moreover, Dict-TTS achieves the comparable objective and subjective PER with the most popular grapheme-to-phoneme tools on two relatively larger datasets (Biaobei and Common Voice (HK)) and maintain a good pronunciation accuracy on a relatively small dataset (JSUT), which demonstrates the superiority of the explicit semantics matching in our S2PA module.

We also compare the pronunciation accuracy of our Dict-TTS with various types of systems, including: 1) a character-based system; 2) a BERT embedding based system [15], where the BERT derived embeddings are concatenated with the character embeddings; 3) NLR [16], a TTS system that directly injects BERT derived knowledge into the linguistic encoder without phoneme labels; 4) Phoneme (G2PM [38]), PortaSpeech with phoneme labels derived from G2PM (a powerful neural G2P system); 5) Phoneme (pypinyin), PortaSpeech with phoneme labels derived from pypinyin (one of the most popular Chinese G2P system). As shown in Table 2, Dict-TTS greatly surpasses the systems that implicitly model the semantic representations for character-to-pronunciation mapping like NLR [16] and shows comparable performance with phoneme-based systems. Since our Dict-TTS does not require any phoneme labels for training, we can pre-train Dict-TTS on a large-scaled ASR dataset [61] with a small amount of effort to improve its generalization capacity. It can be seen that the pronunciation accuracy of Dict-TTS on the Biaobei dataset is significantly improved by pre-training, which demonstrates the effectiveness of our unsupervised polyphone disambiguation framework.

### 4.3 Results of Audio Quality and Prosody

We compare the audio quality, audio prosody, and pitch accuracy[7] of Dict-TTS with the systems evaluated in Subsection 4.2. GT (the ground truth audio) and GT (voc.) (the ground truth audio that is firstly converted into mel-spectrogram and converted back to audio waveform using Hifi-GAN [29]) are also included in this experiment. We keep the text content consistent among different models to exclude other interference factors, only examining the audio quality or prosody. Each audio is listened by at least 20 testers. For audio quality and prosody, we conduct the mean opinion score (MOS) evaluation via Amazon Mechanical Turk.

---

[7]We compute the average dynamic time warping (DTW) [36] distances between the pitch contours of ground-truth speech and synthesized speech.

Table 4: Pronunciation accuracy, audio prosody and audio quality comparisons for ablation study.

| Settings | PER-O | PER-S | SER-S | CMOS-P | CMOS-Q |
|---|---|---|---|---|---|
| Dict-TTS | 2.12% | 1.08% | 6.50% | 0.000 | 0.000 |
| w/o Semantics | 2.15% | 1.13% | 6.50% | -0.280 | -0.135 |
| w/o Gumbel-Softmax | - | 1.19% | 7.75% | -0.014 | -0.021 |

We analyze the MOS in two aspects: MOS-P (Prosody: naturalness of pitch, energy, and duration) and MOS-Q (Quality: clarity, high-frequency, and original timbre reconstruction). We tell the tester to focus on one corresponding aspect and ignore the other aspect when scoring. We put more information about the subjective evaluation in Appendix B.3. As shown in Table 3, for audio quality, Dict-TTS significantly outperforms those TTS systems without phoneme labels and achieves a comparable performance with phoneme-based systems. And for audio prosody and pitch accuracy, Dict-TTS even surpasses the phoneme-based systems, which demonstrates the effectiveness of the extracted semantics from prior dictionary knowledge. We put more analyses on the naturalness of prosody in Appendix E.

Table 3: The audio performance (MOS-Q and MOS-P) and pitch accuracy comparisons. DTW denotes average dynamic time warping distances of pitch in ground-truth and synthesized audio. The mel-spectrograms are converted to waveforms using Hifi-GAN (V3) [29].

| Method | MOS-P | MOS-Q | DTW |
|---|---|---|---|
| GT | 4.48±0.03 | 4.40±0.04 | - |
| GT (voc.) | 4.37±0.04 | 4.26±0.04 | - |
| Character | 3.82±0.08 | 3.88±0.07 | 53.1 |
| BERT Embedding [15] | 3.88±0.07 | 3.63±0.10 | 55.0 |
| NLR [16] | 3.83±0.08 | 3.74±0.08 | 53.3 |
| Phoneme (G2PM) | 3.87±0.08 | 3.90±0.06 | 53.1 |
| Phoneme (pypinyin) | 3.89±0.08 | **3.95±0.06** | 52.6 |
| Dict-TTS | **4.03±0.05** | 3.91±0.04 | **52.4** |

We then visualize the mel-spectrograms generated by the above systems in Figure 5. We can see that Dict-TTS can generate mel-spectrograms with comparable details in harmonics, unvoiced frames, and high-frequency parts with the phoneme-based system, which results in similar natural sounds. Moreover, our Dict-TTS can capture more accurate local changes in pitch and speaking duration, indicating the effectiveness of introducing semantic representations in the dictionary.

## 4.4 Ablation Studies

We conduct ablation studies to demonstrate the effectiveness of designs in Dict-TTS, including the auxiliary semantic information and the Gumbel-Softmax sample strategy. We conduct pronunciation accuracy and CMOS (comparative mean opinion score) evaluations for these ablation studies. The results are shown in Table 4. We can see that CMOS-P drops when we remove the introduced semantic embeddings in Dict-TTS, indicating that the semantic information extracted from the dictionary can improve the audio prosody. Besides, to demonstrate the effectiveness of the Gumbel-Softmax sample strategy, we also compare the weighted sum of the pronunciation embeddings with the Gumbel-Softmax sample strategy. Since measuring PER-O requires one-hot vectors, we do not calculate the PER-O score for the weight-sum version of Dict-TTS. The results are shown in row 3 in Table 4. It can be seen that PER-S and SER-S increase when we use the weighted sum method. In the experiments, the weights of different pronunciations for some characters might be close to each other, which results in relatively worse performance in the subjective results. For example, the two pronunciations "ZH ANG3" and "CH ANG2" of the character "长" might be ambiguous when their weights are close to each other (e.g., $0.6$ and $0.4$). Therefore, to accurately model the subjective pronunciations, we utilize the Gumbel-Softmax function to sample the most likely pronunciation in both training and inference stages.

## 5 Conclusion

In this paper, we proposed Dict-TTS, an unsupervised framework for polyphone disambiguation in end-to-end text-to-speech systems. Dict-TTS uses a semantics-to-pronunciation attention (S2PA) module to explicitly extract the corresponding pronunciations and semantic information from prior dictionary knowledge. The S2PA module can be trained with the end-to-end TTS model with the

guidance of mel-spectrogram reconstruction loss without phoneme labels, which significantly reduces the cost of building a polyphone disambiguation system and further enables the efficient pre-training on large-scaled ASR datasets. Our experimental results in three languages show that Dict-TTS outperforms several strong G2P baseline models in terms of pronunciation accuracy and improves the prosody modeling of the baseline TTS system. Further comprehensive ablation studies verify that each component in Dict-TTS is effective. However, the dictionary knowledge does not contain the tree-structured syntactic information of the input text sequence, which also affects the prosody modeling. Moreover, since we crawl the dictionary from online websites and do not make any specific changes, the performance of Dict-TTS can be further improved by a well-designed dictionary. In the future, we will try to inject syntactic information into Dict-TTS and extend it to more languages.

## 6 Acknowledgments

This work was supported in part by the National Natural Science Foundation of China (Grant No.62072397 and No.61836002), Zhejiang Natural Science Foundation (LR19F020006), Yiwise, and National Key R&D Program of China (Grant No.2020YFC0832505). We appreciate the support from Mindspore, which is a new deep learning computing framework. This work was also supported by Alibaba Group through Alibaba Innovative Research Program.

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
