# OpenReview forum: "Dict-TTS: Learning to Pronounce with Prior Dictionary Knowledge for Text-to-Speech"
_NeurIPS.cc/2022/Conference — NeurIPS 2022 Accept_

### Official Review · Reviewer_nb5D · 2022-07-09

**Rating:** 7
**Confidence:** 5
**Soundness:** 4 excellent
**Presentation:** 3 good
**Contribution:** 4 excellent

**Summary:**

Polyphone disambiguation is a big challenge for end-to-end TTS systems. The paper proposes Dict-TTS, an approach for phonemic disambiguation that can be trained jointly with an end-to-end neural TTS system. The key intuition behind the approach is that semantic context in the input text can be used to select the most appropriate pronunciation from a dictionary. To achieve this goal, the input text is first encoded using a transformer based semantic encoder. The output of this encoder is then used to compute attention weights over the entries in a dictionary using a semantic-to-pronunciation attention module. The embeddings of the most likely pronunciation and the corresponding semantic encoding are fed to the linguistic encoder in the end-to-end TTS system.  The paper shows that DictTTS outperforms character and phoneme baselines (obtained using commonly used grapheme-to-phoneme systems) in terms of pronunciation accuracy in Chinese, Japanese and Cantonese. Pre-training on an ASR dataset can further improve accuracy. The paper also shows that compared to a phoneme-based system, an end-to-end TTS system that incorporates DictTTS achieves similar overall MOS scores and better MOS-prosody scores.


**Questions:**

* The conclusions from the ablation study in Sec 4.4 is not very clear. The paper states that when the top-2 layers of the character based system are removed, the PER/SER increase rapidly but this is not the case for either the phoneme-based system or DictTTS. The paper claims this demonstrates that the Dict-TTS successfully decomposes the character representation, pronunciation, and semantics, which significantly improves the pronunciation accuracy. Why is this the case?
* Why is the attention vector 2-dimensional in Equation 1 i.e. a_{i,1} … a_{i,m} but 3-dimensional on L206 i.e. w_{ij} = \sum_{k=1}^{u} a_{ijk}?
* Did you try an experiment where you used a weighted sum of the embeddings of the pronuncations/semantics vs using the most likely pronunciation?
* Does the dictionary need to be limited to a smaller set of characters for a given input text?
L184 says: "D which contains a sequence of characters C = [c1 , c2 , ..., cn ], where n is the size of characters set in a language" If the language has a large character set, computing an attention weight over the entire dictionary for each input text will be expensive. Do you need any computational shortcuts for the S2PA module to be practical?
* Could you comment whether the noise in the dictionary affects the performance of this approach?

* typo: L21/92 'there' -> 'they'



**Limitations:**

The authors have discussed some limitations in Sec 5. The authors have also discussed potential negative societal impact of their work.

**Strengths And Weaknesses:**

Strengths:
* Proposes a new approach for polyphone disambiguation which can be jointly trained with the end-to-end TTS model using mel-spectrogram reconstruction loss without requiring phonemic labels, which are expensive to obtain.
* Performance of the system improves by pre-training on an ASR corpus.
* The approach yield improvements in pronunciation accuracy on three languages and improves in prosody of the underlying TTS system

Weaknesses:
* Some parts of the paper are hard to follow (see details below)

Update: In their revisions, the authors have adequately addressed my concerns.

---

> ### Author Response · Authors · 2022-08-02
> **Response to Reviewer nb5D**
>
> We are grateful for your positive review and valuable feedback, and we hope our response fully resolves your concern.
>
>
>
> **[About the ablation study in Section 4.4] (Question 1)**
>
> We apologize for the confusing ablation studies in Section 4.4. We have conducted new experiments to demonstrate the effectiveness of  designs in Dict-TTS, including the auxiliary semantic information and the Gumbel-Softmax sample strategy. More details can be found in Section 4.4 in the revised version of the paper. Thanks for the reviewer’s kind and helpful comments!
>
>
>
> **[About the confusing 2-dimensional terms $a_{i,1}$ in Equation 1] (Question2)**
>
> The attention vector $a$ is a 3-dimensional vector. We are sorry for our confusing terms in Section 3.3. We have clarified these terms and marked them blue in the revised version of the paper.
>
>
>
> **[About the experiment of a weighted sum of the pronunciations embeddings/semantics vs using the most likely pronunciation] (Question 3)**
>
> Yes, we have conducted this experiment in the Biaobei dataset. Since measuring PER-O requires one-hot vectors, we do not calculate the PER-O score for the weight-sum version of Dict-TTS. The results are shown in the following table:
>
> | Methods                   | PER-O | PER-S | SER-S |
> | ------------------------- | ----- | ----- | ----- |
> | Dict-TTS (Weighted Sum)   | - | 1.19% | 7.75% |
> | Dict-TTS (Gumble Softmax) | 2.12% | 1.08% | 6.50% |
>
> It can be seen that PER-S and SER-S increase when we use the weighted sum method. In the experiments, the weights of different pronunciations for some characters might be close to each other, which results in relatively worse performance in the subjective results. For example, the two pronunciations "ZH ANG3" and "CH ANG2" of the character ``长'' might be ambiguous when their weights are close to each other (e.g., $0.6$ and $0.4$). Therefore, to accurately model the subjective pronunciations, we utilize the Gumbel-Softmax function to sample the most likely pronunciation in both training and inference stages.
>
>
>
> **[About the set of characters in the dictionary and computational shortcuts for the S2PA module] (Question 4)**
>
> No, the dictionary does not need to be limited to a smaller set of characters for a given input text. For example, we use the full set of characters in the Chinese dictionary obtained from https://github.com/yihui/zdict, which contains 7030 characters.
>
> Yes, computing an attention weight over the entire dictionary for each input text is quite expensive. As shown in Section 4.1, Line 247-249, for computational efficiency, we firstly use the pre-trained XLM-R model to extract the semantic representations from the raw text of the whole dictionary and record them on the disk. We then load the mini-batch along with the pre-constructed dictionary representation during training and testing. Besides, we also restrict the length of the dictionary entry $e_{i,j}$ to be less than 50 characters for computational efficiency.
>
> As shown in Section 4.1, Line 257, we use a batch size of 40 sentences in Dict-TTS training. And we a use batch size of 64 sentences in all baseline systems (following PortaSpeech [1]). We are sure that the memory usage and total training time are consistent among all experiments.
>
> | Dictionary entry length | Entry number |
> | ----------------------- | ------------ |
> | 0<=x<50                 | 5193         |
> | 50<=x<100               | 2042         |
> | x>=100                  | 956         |
>
>
>
> **[About whether the noise in the dictionary affects the performance] (Question 5)**
>
> We are sorry that we may not understand the meaning of "noise in the dictionary". Could you please clarify it so that we can make a better response?
>
> According to our comprehensions, "noise in the dictionary" means the wrong or inappropriate definitions or usages in the dictionary. However, the dictionaries used in our experiments have been adequately revised in history, which rarely have wrong definitions or usage.
>
>
>
> **[About the typo: L21/92 'there' -> 'they'] (Question 6)**
>
> We think that "There" in Line 21 and Line 92 are not typos.
>
>
>
> Again, we appreciate your positive review and hope our response can fully resolve your concerns.
>
>
>
> **[References]**
>
> [1] Yi Ren, Jinglin Liu, and Zhou Zhao. Portaspeech: Portable and high-quality generative text-to-speech. Advances in Neural Information Processing Systems, 34, 2021.

---

> > ### Comment · Reviewer_nb5D · 2022-08-09
> > **Satisfied with the response**
> >
> > Thanks to the authors for revising the paper. The response answers my questions. I am updating the score accordingly.

---

### Official Review · Reviewer_ksPw · 2022-07-11

**Rating:** 4
**Confidence:** 5
**Soundness:** 1 poor
**Presentation:** 2 fair
**Contribution:** 1 poor

**Summary:**

This work presents a method to increase the pronunciation accuracy of a speech synthesis system without phoneme labels by using an online dictionary.

**Questions:**

The authors need to clarify the limitations of the method presented and the points described above.

**Limitations:**

Comments on the limitations of the work are mentioned above. \
There is no negative societal impact.

**Strengths And Weaknesses:**

## Strengths
1. High intelligibility can be obtained without phoneme label in Text-to-Speech task using logogram.\

## Weaknesses
1. The authors claim to present a method to solve the polyphone disambiguation problem, which is a problem in logograms such as Chinese, but is less problematic in phonograms such as English. The authors argue in this section that the characteristics of the logographic writing system can be easily extended to alphabetic languages (e.g., English) by replacing “character” with “word”, but this seems to be an inappropriate explanation that does not take into account that the phonogram basically displays the pronunciation unlike the logogram.

1. The authors mention the loss of information caused by converting grapheme to phoneme in comparison with “Phoneme-based TTS systems”, but it is not related to conversion errors that occur during the conversion process to phoneme (An error in which Pinyin is converted differently from context or semantic meaning.), which the authors tried to solve. The problem of information loss in the conversion process to phoneme has already been explored in previous work such as [1] and [2], and this comparison is not appropriate because it is not the same as the problem claimed in this work.

1. In addition, the authors excluded the space token from the comparison between “I scream” and “Icecream”, this is an inappropriate example given that unlike Chinese which the space token is not commonly used, the space token is information that greatly influences prosody prediction in English.

1. The method presented in section 3.3 does not seem to be effective considering that the dictionary is composed of word units and that one grapheme can be mapped to multiple phonemes in many cases in phonograms(e.g., grapheme ‘e’ in English). The authors need to explain the limitations of the presented method and modify the scope of the claims.

1. The authors claim that the character representations are well distributed in the semantic space. To support this, the authors need to prove how the semantic space is defined and well distributed. It is not appropriate to claim that the character representations are well distributed in the semantic space based solely on predicting pronunciation well.

1. Referring to Table 1, in the case of Japanese, the performance is significantly lower than when using the open source G2P module, and the authors claim this as “which demonstrates the superiority of the explicit semantics matching in our S2PA module.”.
It is difficult to understand how this result demonstrates superiority.
In addition, the authors assert in the Conclusion as follows:
“Our experimental results in three languages show that Dict-TTS outperforms several strong G2P baseline models in terms of pronunciation accuracy and improves the prosody modeling of the baseline TTS system.”
Clearly, it showed better performance for only two languages and a significant performance degradation for Japanese, so I think this part also has a problem with the argument, and I have doubts about the overall completeness and reliability of the paper.

1. In terms of generalization, “Biaobei” and “HK” are composed entirely of logograms, and “JSUT” is different in that it is a mixture of phonograms and logograms. According to the experimental results presented by the authors, the proposed method has the potential to work only in the logograms, can be viewed as a solution that operates under specific conditions, and is not of high significance compared to solutions that can be generalized.


[1] Jia, Ye, et al. "PnG BERT: Augmented BERT on phonemes and graphemes for neural TTS." arXiv preprint arXiv:2103.15060 (2021).\
[2] Kastner, Kyle, et al. "Representation mixing for TTS synthesis." ICASSP 2019-2019 IEEE International Conference on Acoustics, Speech and Signal Processing (ICASSP). IEEE, 2019.

---

> ### Author Response · Authors · 2022-08-02
> **Response to Reviewer ksPw**
>
> We thank the reviewer for the constructive feedback and for considering our work as "High intelligibility can be obtained without phoneme label in Text-to-Speech task using logogram". We understand that your concerns are mainly related to the paper’s generalization limitations and claims. We hope our response resolves your concerns fully.
>
>
> **[About question 1]**
>
> Yes, we agree that the polyphone disambiguation problem is an important problem in logograms such as Chinese, but is less problematic in phonograms such as English. Although there are fewer polyphones and heteronyms in phonograms, the pronunciations of the polyphones in phonograms should also be deduced based on the semantic contexts. For example, "resume" in English can be pronounced as "[ri′zju:m]" (means to go on or continue after interruption) or "[rezjumei]" (means curriculum vitae). Therefore, Dict-TTS can also disambiguate the polyphones in phonograms by replacing "character" with "word". Our methods can also be used as the modules to retrieve the correct pronunciation for polyphones and heteronyms in English G2P process (e.g., the Algorithm step 2 in https://github.com/Kyubyong/g2p).
>
>
>
> **[About question 2]**
>
> The main problem our Dict-TTS tries to solve is polyphone disambiguation in TTS systems and the loss of information caused by G2P pre-processing in phoneme-based TTS systems is only a part of our preliminary analyses. Section 3.2 and Section 3.3 can be summarized as follows:
>
> 1. We make preliminary analyses about the challenges and information loss faced in character-based and phoneme-based TTS systems.
> 2. Based on these analyses, we propose our Dict-TTS. In Dict-TTS, we firstly capture the semantic information of the input character sequence. Then the S2PA module deduce the pronunciations based on the extracted semantic information and the dictionary knowledge. Finally we obtain the expressive pronunciation hidden states from the deduced pronunciations and semantic information.
>
> Moreover, although the problem of information loss in the conversion process to phoneme has already been explored in previous work such as [1] and [2], their methods require  (phoneme + character) or (phoneme + grapheme + word-level alignment) as input features. Our work aims at modeling natural pronunciations based on the input character sequence and dictionary knowledge, which is a different setting.
>
>
>
> **[About question 3]**
>
> Thanks for your advice! We agree that the space token is information that greatly influences prosody prediction in English and our example is inappropriate. We have changed the example of "I scream | Icecream" to "whether | weather" and marked them blue in the revised version of the paper.
>
>
>
> **[About question 4]**
>
> Yes, we agree that one grapheme can be mapped to multiple phonemes in many cases in phonograms (e.g., grapheme ‘e’ in English). **However, our work aims at polyphone disambiguation problem in G2P conversion.**
>
> For alphabetic languages like English, lexicon cannot cover the pronunciations of all the words. Thus, the G2P conversion for English is mainly responsible for generating the pronunciations of out-of-vocabulary words [3]. Although the polyphone disambiguation is less problematic in these languages, our methods can still be used as the modules to retrieve the correct pronunciation for polyphones and heteronyms in their G2P process (e.g., the Algorithm step 2 in https://github.com/Kyubyong/g2p). By replacing "character" with "word" in Section 3.3, Dict-TTS is effective in the polyphone or heteronym disambiguation problem in alphabetic languages.
>
> For logograms like Chinese, although the lexicon can cover nearly all the characters, there are a lot of polyphones that can only be decided according to the context of a character. Thus, G2P conversion in this kind of languages is mainly responsible for polyphone disambiguation, which decides the appropriate pronunciation based on the current word context. Therefore, polyphone disambiguation is crucial in these languages and our method is an effective solution for polyphone disambiguation problem.
>
> Thanks for your suggestions. We have explained how substantial a problem this is for a variety of languages in Appendix G and marked them blue in the revised version of the paper.
>
>
>
> **[About question 5]**
>
> At first, we are sure that the dictionary embeddings extracted by the pre-trained language model are well distributed in the semantic space. High pronunciation accuracy in the experiments means the pronunciation weights (the semantic similarities) in our S2PA module are accurate enough. It also demonstrates the high similarities between the dictionary embeddings extracted by the pre-trained language model and the character-level representations extracted by our semantics encoder. Therefore, we conclude that the character representations are well distributed in the semantic space.

---

> > ### Author Response · Authors · 2022-08-02
> > **Due to the limited number of characters, we show the rest of our reply here.**
> >
> >
> >
> > **[About question 6 and 7]**
> >
> > **Our work aims at the polyphone disambiguation problem in G2P conversion.**
> >
> > Yes, "JSUT" is a mixture of phonograms and logograms, which is different from "Biaobei" and "Common Voice (HK)". Japanese writing system consists of two types of characters: the syllabic kana – hiragana (平仮名) and katakana (片仮名) – and kanji (漢字). In our analysis, 32.42% of the characters in JSUT dataset are kanji. The pronunciations of a part of the kanji can not only be specified by the semantic information and should be specified by empirical pronunciation rules. For example, most kanji (漢字) can be pronounced multiple ways: **on-yomi (音読み)** and **kun-yomi (訓読み)**. Although the compound kanji usually uses on-yomi and one kanji probably uses kunyomi, the different readings are largely just chosen empirically in practice. Our Dict-TTS has the potential to work only for the kanji whose pronunciation should be specified based on the semantic meaning. Due to the characteristics of Japanese writing systems, in Table 1, although Dict-TTS surpasses the character-based system, it does not show comparable performance with the open source G2P module in Japanese. But as shown in Section 3.4 Line 226, our method is compatible with the predefined rules from language experts by directly adding specific rules to pronunciation weight. We are sure that the performance of our method can be further improved by introducing the pronunciation rules in Japanese (e.g., the rules in the rule-based G2P baseline "pyopenjtalk").
> >
> > All in all, our work aims at the polyphone disambiguation problem in G2P conversion and can be generalized to various languages. But the polyphone disambiguation problem may be less problematic in some languages. Thanks for your suggestions. We have clarified these limitations in Appendix G in the revised version of the paper.
> >
> >
> >
> > Finally, we appreciate the reviewer’s valuable reviews and believe some misunderstandings are due to our clarity. Hope our response can address your concerns.
> >
> >
> > **[References]**
> >
> > [1] Jia, Ye, et al. "PnG BERT: Augmented BERT on phonemes and graphemes for neural TTS." arXiv preprint arXiv:2103.15060 (2021).
> >
> > [2] Kastner, Kyle, et al. "Representation mixing for TTS synthesis." ICASSP 2019-2019 IEEE International Conference on Acoustics, Speech and Signal Processing (ICASSP). IEEE, 2019.
> >
> > [3] Tan, Xu, et al. "A survey on neural speech synthesis." *arXiv preprint arXiv:2106.15561* (2021).

---

### Official Review · Reviewer_GMo7 · 2022-07-11

**Rating:** 7
**Confidence:** 4
**Soundness:** 3 good
**Presentation:** 3 good
**Contribution:** 3 good

**Summary:**

This paper describes an approach to leverage available human readable dictionaries to help improve TTS pronunciation modeling, specifically for polyphonous lexical items.  The aim here is to leverage the semantic information from the definition and example to guide selection of an appropriation pronunciations.

**Questions:**

* It would be useful to demonstrate how substantial a problem this is for a variety of languages.  How necessary is solving this problem for delivering high quality TTS?  Some additional motivation to this end could be helpful.

* In section 3.3. how much preprocessing of the dictionary text is important here?  e.g. is the decomposition into "Character" "Pronunciation" "Definitions" "Usages" always used?  Are there ever errors to this decomposition?

* In Section 3.3 it is claimed that "..the model can easily deduce the correct pronunciation and semantics based on the lexicon knowledge like human brain".  In what way is this like the human brain's processing?  Would any function that uses a "semantic" embedding to disambiguate pronunciation be "like the human brain"?

* Section 3.4 Do you have any insight into the quality of the pronunciations obtained from the "low-quality text-speech pairs" obtained from ASR data?  E.g. Librispeech is divided into "clean" and "other" partitions based on how easy an ASR model can recognize them.  It would be interesting to understand the quality/quantity tradeoffs that are made in this work.

* Table 1 - how much should PER-O be trusted as a gold standard? In some languages there are multiple valid pronunciations of worse e.g. English "the" pronounced as /ðə/ vs. /ðiː/.  The example pronunciations are quite different, but to what degree is some pronunciation variation tolerated?

* Table 2 A comparison to PNG-BERT would make sense here since it is another approach to include semantics in the TTS frontend.  Is there a reason this is not used?

* Table 3: is the Dict-TTS entry pretrained or not?

* Table 3: is there an explanation for why the quality score is impacted by the pronunciation representation?  It is particularly interesting that the prosody score is positively impacted by the improved pronunciation modeling of Dict-TTS. How much interaction if any is there between pronunciation, lexical tone, and subjective prosody responses here?

* Table 3: the DTW measure would be impacted by both lexical tone and prosodic realization. Is there any effort to disentangle these when evaluating the impact of these approaches?

* Table 4, Section 4.4. This ablation studey of removing two layers is somewhat strange.  Without retraining the model with fewer layers, there isn't an expectation that useful information would be available from lower layers to be used by a higher layer in a network.

**Limitations:**

Some limitations of the work are discussed.  The potential negative societal impact of the work is not addressed.

**Strengths And Weaknesses:**

Strengths
* Clear motivation, mostly very well described technical approach.
* Good improvement over comparable studies.
* Interesting use of distant or weak supervision for this task.

Weaknesses
* The problem being solved may be somewhat narrow, though important for TTS.
* The ablation study of removing network layers is not altogether convincing.

---

> ### Author Response · Authors · 2022-08-02
> **Response to Reviewer GMo7**
>
> Thanks for your positive review and valuable comments, and we hope our response fully resolves your concerns.
>
> **[About how substantial a problem this is for a variety of languages] (Question 1)**
>
> The polyphone disambiguation problem is critical in logographic languages such as Chinese, but is less problematic in phonograms like English.
>
> For logographic languages like Chinese, although the lexicon can cover nearly all the characters, there are a lot of polyphones that can only be decided according to the context of a character. Thus, G2P conversion in this kind of languages is mainly responsible for polyphone disambiguation, which decides the appropriate pronunciation based on the current word context. Therefore, polyphone disambiguation is crucial in these languages and our method is an effective solution for the polyphone disambiguation problem.
>
> For alphabetic languages like English, lexicon cannot cover the pronunciations of all the words. Thus, the G2P conversion for English is mainly responsible for generating the pronunciations of out-of-vocabulary words [1]. Although the polyphone disambiguation is less problematic in these languages, our methods can still be used as the modules to retrieve the correct pronunciation for polyphones and heteronyms in their G2P process (e.g., the Algorithm step 2 in https://github.com/Kyubyong/g2p).
>
> Thanks for your helpful suggestions. We have explained how substantial a problem this is for a variety of languages in Appendix G and marked them blue in the revised version of the paper.
>
>
>
> **[About the pre-processing the dictionary text] (Question 2)**
>
> **Question:** how much pre-processing of the dictionary text is important here? e.g., is the decomposition into "Character" "Pronunciation" "Definitions" "Usages" always used?
>
> **Answer:** In our experiments, we just crawl the dictionaries and do not need too much pre-processing. Besides, the decomposition into "Character" "pronunciation" "Definitions" "Usages" is not necessary, but the decomposition into "Character" "pronunciation" "Definitions or Usages" is necessary.
>
> **Question:** Are there ever errors to this decomposition?
>
> **Answer:** There are rarely errors to this decomposition since the online dictionaries used in people's daily life are already well-organized and decomposed (e.g., the online dictionaries listed in Appendix A.1).
>
>
>
> **[About the way like the human brain's processing] (Question 3)**
>
> In Section 1, Line 37-39, "When one is confused about the acoustic pronunciation of a specific polyphone, he or she will resort to the dictionary website to infer its exact reading based on the semantic context". In our S2PA module, the semantic encoder aims at comprehending the semantic contexts in the input character sequence. The semantic similarity between the input character representations and the dictionary entries is measured for deducing the correct pronunciations. Therefore, we claim that "... the model can easily deduce the correct pronunciation and semantics based on the lexicon knowledge like human brain".
>
>
>
> **[About the insight into the quality of the pronunciations obtained from the "low-quality text-speech pairs] (Question 4)**
>
> In our experiments, we use the Wenetspeech dataset [2] for Dict-TTS (pre-trained). The Wenetspeech dataset contains 10005 hours of text-speech pairs with 0.95 ~ 1.0 confidence. We use the 1000 hours "M training subset" with 1.0 confidence for our Dict-TTS pre-training. Among the M subset, there are approximately 400 hours of audio from podcasts which can be seen as the "clean" partition and 600 hours of audio from Youtube which can be seen as the "other" partition. Most of the audio samples in the "Youtube" set come from online dramas, which contains various background music or loud noise. Besides, the pronunciations in these audio samples may not be accurate enough. We have pre-trained our Dict-TTS with different subsets in the WenetSpeech dataset and the results are shown in the following table. It can be seen that although the audio quantity in the "Youtube" set and "Podcast + Youtube" set are larger, the pronunciation accuracy of Dict-TTS (pre-trained) is negatively impacted by the poor audio quality.
>
> | Methods                | Set for pre-training  | PER-O     | PER-S     | SER-S     |
> | ---------------------- | --------------------- | --------- | --------- | --------- |
> | Dict-TTS (pre-trained) | Podcast Set           | **1.54%** | **0.79%** | **4.25%** |
> | Dict-TTS (pre-trained) | Youtube Set           | 1.97%     | 1.02%     | 6.25%     |
> | Dict-TTS (pre-trained) | Podcast + Youtube Set | 1.63%     | 0.87%     | 4.75%     |
> | Dict-TTS               | None                  | 2.12%     | 1.08%     | 6.50%     |

---

> > ### Author Response · Authors · 2022-08-02
> > **Due to the limited number of characters, we show the rest of our reply here.**
> >
> >
> > **[About how much should PER-O be trusted as a gold standard]  (Question 5)**
> >
> > Yes, in some languages, there are multiple valid pronunciations of worse. However, for example, the gold pronunciation of the word "the" depends on the first sound of the word that comes after it, which can be called **pronunciation rules**. In our experiments, the Mandarin character "一" also has the sandhi pronunciation rule, e.g. "一" before tone4 should be "yi2" (一段) and when "一" is an ordinal word, it should be "yi1".
> >
> > We are sure that the ground truth labels used in the PER-O experiments conform to the pronunciation rules in those languages and can be trusted as a gold standard.
> >
> >
> >
> > **[About the comparison to PNG-BERT] (Question 6)**
> >
> > 1. PNG-BERT [3] has not released its code officially. We find an unofficial implementation (https://github.com/ishine/PnG-BERT), but we do not obtain satisfying results.
> > 2. The basic architecture of PNG-BERT [3] is Non-attentive Tacotron [4], which is quite different from Portaspeech [5] (the baseline system used in our experiments).
> > 3. PNG-BERT [3] requires phoneme, character, and word-level alignment as input features. However, our work aims at modeling natural pronunciations based on the input character sequence and dictionary knowledge, which is a different setting.
> >
> > Therefore, for fair comparisons, we do not use PNG-BERT [3] as one of the baseline systems in our experiments.
> >
> >
> >
> > **[About the question "Table 3: is the Dict-TTS entry pretrained or not?" ] (Question 7)**
> >
> > For fair comparisons, the Dict-TTS entry in Table 3 is not pretrained.
> >
> >
> >
> > **[About the impacted quality score and the interaction between pronunciation, lexical tone, and subjective prosody responses in Table 3] (Question 8)**
> >
> > **Question:** Is there an explanation for why the quality score is impacted by the pronunciation representation?
> >
> > **Answer:** For audio quality evaluation, we tell listeners to "focus on examining the naturalness of audio quality (e.g., noise, timbre, sound clarity, and high-frequency details)". The pronunciation accuracy will influence the sound clarity in audio quality evaluation.
> >
> > **Question:** How much interaction if any is there between pronunciation, lexical tone, and subjective prosody responses here?
> >
> > **Answer:** For subjective prosody evaluations, we focus on examining pitch, energy, and duration. The pronunciation and lexical tone may affect the local pitch trajectory and energy distribution in subjective prosody evaluation. But as shown in the ablation studies in Section 4.4, the improvement in pronunciation modeling of Dict-TTS is mainly due to the semantic information extracted from the dictionary.
> >
> >
> > **[About disentangling the lexical tone and prosodic realization in evaluations] (Question 9)**
> >
> > The lexical tone can be evaluated by the PER-O, PER-S, and SER-S metrics. The prosodic realization can be evaluated by the duration errors and character-level average pitch errors. For duration errors, we calculate the character-level duration MSE. For character-level average pitch errors, we firstly calculate the mean pitch for each character's region in the mel spectrogram according to the Montreal Forced Aligner (MFA) to remove the influence of lexical tone, and then we calculate the MSE of the mean pitch sequences. We present the results on the Biaobei dataset in the following tables:
> >
> > | Method             | Duration Error (ms) | Pitch error |
> > | ------------------ | ------------------- | ----------- |
> > | Character          | 36.2                | 1424.6      |
> > | BERT Embedding     | 35.7                | 1312.1      |
> > | NLR                | 36.4                | 1414.3      |
> > | Phoneme (G2PM)     | 35.8                | 1341.7      |
> > | Phoneme (pypinyin) | 35.3                | 1308.8      |
> > | Dict-TTS           | 34.4                | 1232.3      |
> >
> > We have attached these results to Appendix E and marked them blue in the new version of the paper.
> >
> >
> > **[About the ablation studies in Section 4.4] (Question 10)**
> >
> > We apologize for the confusing ablation studies in Section 4.4. We have conducted new experiments to demonstrate the effectiveness of  designs in Dict-TTS, including the auxiliary semantic information and the Gumbel-Softmax sample strategy. More details can be found in Section 4.4 in the revised version of the paper. Thanks for the reviewer’s kind and helpful suggestions!
> >
> >
> > Again, we thank the reviewer for the insightful reviews and “Accept” recommendation for our paper.
> >
> > **[References]**
> >
> > [1] Tan, Xu, et al. "A survey on neural speech synthesis." 2021.
> >
> > [2] Binbin Zhang, et al. Wenetspeech: A 10000+ hours multi-domain mandarin corpus for speech recognition. 2022.
> >
> > [3] Jia, Ye, et al. "PnG BERT: Augmented BERT on phonemes and graphemes for neural TTS." 2021.
> >
> > [4] J. Shen, et al. “Non-Attentive Tacotron: Robust and controllable neural TTS synthesis including unsupervised duration modeling,” 2020.
> >
> > [5] Yi Ren, et al. Portaspeech: Portable and high-quality generative text-to-speech. 2021.

---

### Official Review · Reviewer_mE8x · 2022-07-12

**Rating:** 7
**Confidence:** 4
**Soundness:** 3 good
**Presentation:** 3 good
**Contribution:** 3 good

**Summary:**

The paper proposes Dict-TTS that can infer the corresponding pronunciations for the given input text sequence by incorporating the prior information from the online website dictionary.  For deriving the pronunciations, the paper proposes a semantics-to-pronunciation attention (S2PA) module which finds the correct pronunciations by matching the semantic information between the input text sequence and the dictionary entries.  The proposed S2PA module can be incorporated into the end-to-end TTS system and can be trained simultaneously with the TTS system using the mel-spectrogram reconstruction loss.  The idea is interesting and is validated by extensive experiments on three datasets with different languages.

**Questions:**

1) In Section 4.4 “Ablation Studies”, the paper claims that “Dict-TTS successfully decompose the character representation, pronunciation, and semantics, which significantly improves the pronunciation accuracy.”  How could this conclusion be drawn from the ablation study experiments?  For the proposed Dict-TTS, removing the top two layers of the linguistic encoder only increases the PER and SER slightly.  But do the results indicate the success of decomposition (of character representation, pronunciation, and semantics)?  The paper should give more explanations about this, or should prevent over-claiming.

2) Also in Section 4.4, the paper reports the results by removing the top two layers of the linguistic encoder for different systems including the character-based, phoneme-based and the Dict-TTS.  Why are these top two layers of the linguistic encoder important?  What about the other layers of the linguistic encoder?

3) In Section 3.4, it is said that “our method are compatible with the predefined results … by directly adding specific rules to pronunciation weight $w_{i,j}$.  It would be interesting to elaborate more on how to add the rules to the pronunciation weight.

4) In Section 3.3, for a pronunciation $p_{i,j}$, it may corresponds to a dictionary entry $e_{i,j}$ with several different items including different definition examples and different usage examples, etc.  For example, in Figure 1, the first pronunciation corresponds to two definition examples, and seven or eight usage examples.  Are these examples merged together as a single character sequence, leading to $[e_{i,j,1}, …, e_{i,j,u}]$ with $u$ characters?  The authors should give more details about this design.  Will the operation of merging different examples into a single character sequence affect the performance?

5) In Section 3.3, some terms might be confusing and need be clarified.

* Line 202, $a_{i,1}$ should be $a_{i,j}$?

* Line 202, $c_i$ might be confused with the character $c_i$ in Dictionary (i.e. $c_i$ in Line 186).  The $c_i$ in Figure 4 should also be clarified.

* Line 206, it is not clear what does the term $a_{i,j,k}$ mean.


**Limitations:**

Yes, the authors address the impacts and limitations in Appendix E.

**Strengths And Weaknesses:**

Strengths:

1) The idea of using prior knowledge from the dictionary is interesting, and the method of using S2PA module is novel.  Specifically, the S2PA module incorporates the semantic information for polyphone disambiguation, which imitates the “dictionary lookup” practice in human daily life.

2) The proposed method for polyphone disambiguation (grapheme to phoneme) can be trained in an unsupervised manner and can be trained simultaneously with the TTS modules in an end-to-end manner.  The method greatly eases the process for building a TTS system which directly accepts the raw text as input (i.e. character-based TTS system).

3) The proposed method also provides the possibility to pre-train the model on large-scale ASR dataset to improve the generalization capacity for improving the polyphone disambiguation performance.

Weaknesses:

My main concerns are mainly related to the experiments.  Please refer to the following Questions section for details.

---

> ### Author Response · Authors · 2022-08-02
> **Response to Reviewer mE8x**
>
> We are grateful for your positive review and valuable comments, and we hope our response fully resolves your concerns.
>
>
>
> **[About the ablation studies in Section 4.4] (Question 1 and Question 2)**
>
> We apologize for the confusing ablation studies in Section 4.4. We have conducted new experiments to demonstrate the effectiveness of  designs in Dict-TTS, including the auxiliary semantic information and the Gumbel-Softmax sample strategy. More details can be found in Section 4.4 in the revised version of the paper. Thanks for the reviewer’s kind and helpful suggestions!
>
>
>
> **[About how to add the rules to the pronunciation weight] (Question 3)**
>
> In Mandarin, there are some pronunciation rules (like "sandhi rules") that can not be learned from the dictionary. For example,  "一" before tone4 should be "yi2" (e.g., "一段") and when "一" is an ordinal word, it should be "yi1" (e.g., "一四九五年"). According to these pronunciation rules, we can obtain the correct pronunciation labels for some specific characters based on the input character sequence's part-of-speech (POS) tags. After we obtain the correct pronunciation labels for these specific characters, we can directly force the pronunciation weights of these characters to be the ground truth values.
>
> And in our experiments for Mandarin, we only use the sandhi rules from the PaddleSpeech frontend (https://github.com/PaddlePaddle/PaddleSpeech/blob/develop/paddlespeech/t2s/frontend/tone_sandhi.py) for Dict-TTS and phoneme-based baseline systems (pypinyin and G2PM). Although we can use a portion of the rules in the rule-based baselines (e.g., pypinyin) to further improve Dict-TTS's PER and SER, for a fair comparison, we only add the sandhi rules from PaddleSpeech to the pronunciation weights of our Dict-TTS. We have attached these explanations to Appendix F in the new version of the paper.
>
>
>
> **[About more details about the dictionary design] (Question 4)**
>
> Thanks for the reviewer’s feedback that requests more details about the dictionary design. We merge the definition examples and usage examples as a single character sequence. We have added more details about this design in Section 3.3 and marked them blue in the revised version of the paper.
>
> Our method does not require them to be carefully structured like the dictionary shown in Figure 1. For example, as shown in Figure 2, some characters' pronunciations in the Chinese dictionary used in the experiments (https://github.com/yihui/zdict) may only have several usage examples. And usage examples of the pronunciation "L E4" are similar in terms of semantics. Therefore, the operation of merging different examples into a single character sequence will not affect the performance of semantics matching.
>
>
>
> **[About some confusing terms] (Question 5)**
>
> We are sorry for the confusing terms in Section 3.3. We have clarified these terms and marked them blue in the revised version of the paper.
>
>
>
> Again, we thank the reviewer for the insightful review and “Accept” recommendation for our paper.

---

### Author Response · Authors · 2022-08-02
**Summary of the Rebuttal Revision**

## Summary of the rebuttal revision

We would like to thank the reviewers for their constructive reviews! Here we summarize the revision of the manuscript according to the comments and suggestions of reviewers:

- In section 3.3, we clarified some terms and modified the description for our S2PA module.
- In section 4.4, we conducted new experiments for Dict-TTS to demonstrate the effectiveness of auxiliary semantic information and the Gumbel-Softmax sample strategy.
- In Appendix E in the supplementary material, we further analyzed the naturalness of prosody and rhythm for different TTS systems.
- In Appendix F in the supplementary material, we introduced how to add Rules to the pronunciation weights.
- In Appendix G in the supplementary material, we describe the importance of polyphone disambiguation for various languages.

---

### Author Response · Authors · 2022-08-07
**Dear AC and reviewers,**

Thanks again for your great efforts and valuable comments.

We have carefully addressed the main concerns and provided detailed responses to each reviewer. We hope you might find the responses satisfactory. As the end of the rebuttal phase is approaching, we would be grateful if we could hear your feedback regarding our answers to the reviews. We will be very happy to clarify any remaining points (if any).

Thanks in advance,
Paper 2575 authors

---

### Public Comment · Authors · 2023-10-19
**Adding additional explanation about Neural Lexicon Reader**

The authors have added an additional explanation about Neural Lexicon Reader (arXiv:2110.09698), which is a concurrent work that also proposed to resolve polyphones in end-to-end TTS by extracting implicit pronunciation information from relevant dictionary texts encoded by XLM-R, and the updated version of the paper has been uploaded to arXiv.

---

### Meta-Review · Area_Chair_Svep · 2022-08-26

**Recommendation:** Accept
**Confidence:** Certain

**Metareview:**

The reviewers generally liked the proposed approach in this paper, agreed that it is novel, and that the experiments showed good improvements over reasonable baselines. There was broad concern about the ablation study in the original paper (one shared by the AC), but the authors revised that section during the discussion period to the satisfaction of three of the reviewers. While three reviewers recommend that the paper be accepted, one reviewer recommends a borderline reject. The reviewer stuck to this recommendation after the discussion period, primarily citing concerns about whether or not the method is broadly applicable versus being limited primarily to being useful for logographic languages. While I am recommending that this paper be accepted, I urge the authors to expand their discussion of the limitations of the method in Appendix G. I think the discussion with reviewer ksPw of the JSUT results and the fact that Japanese writing comprises both more alphabetic and more logographic elements would be a valuable addition to that appendix and would help to clarify the contributions and limitations of the proposed method.


**Award:**

No

---

### Decision · Program_Chairs · 2022-09-14

Accept